# Intestinal Microbiota, Anti-Inflammatory, and Anti-Oxidative Status of Broiler Chickens Fed Diets Containing Mushroom Waste Compost By-Products

**DOI:** 10.3390/ani11092550

**Published:** 2021-08-30

**Authors:** Wen Yang Chuang, Li Jen Lin, Hsin Der Shih, Yih Min Shy, Shang Chang Chang, Tzu Tai Lee

**Affiliations:** 1Department of Animal Science, National Chung Hsing University, Taichung 404, Taiwan; xssaazxssaaz@yahoo.com.tw; 2School of Chinese Medicine, College of Chinese Medicine, China Medical University, Taichung 404, Taiwan; linlijen@mail.cmu.edu.tw; 3Taiwan Agricultural Research Institute, Council of Agriculture, Executive Yuan, Taichung 413, Taiwan; tedshih@tari.gov.tw; 4Hsinchu Branch, Livestock Research Institute, Council of Agriculture, Miaoli 368, Taiwan; emshy@tlri.gov.tw; 5Kaohsiung Animal Propagation Station, Livestock Research Institute, Council of Agriculture, Pingtung 912, Taiwan; macawh@tlri.gov.tw; 6The iEGG and Animal Biotechnology Center, National Chung Hsing University, Taichung 402, Taiwan

**Keywords:** agriculture by-product, anti-inflammatory, antioxidant, broiler, microbiota

## Abstract

**Simple Summary:**

This study investigated the effects of *Pennisetum purpureum* waste mushroom compost (PWMC) supplementation on microbiota, as well as its effects on the antioxidant capacities and inflammatory response characteristics of broiler chickens. Results showed that a 5% replacement of a soybean meal via PWMC feeds could enhance the health of chickens by maintaining intestinal microbiota balance, improving antioxidant capacities, and decreasing inflammatory response. Supplementation also further increased the appetite of broilers, thereby improving their growth performances. Furthermore, the number of *Lactobacillus* also increased in the intestinal tracts. High-fiber mushroom waste compost effectively increased the mRNA expression of appetite-related genes in broilers. The broilers’ gut barrier function also increased, while the number of *Turicibacter* in the cecum decreased. It was concluded that a 5% replacement of a soybean meal via PWMC could enhance intestinal health; therefore, it is recommended for the broiler chickens’ diet.

**Abstract:**

This study investigated the effects of using mushroom waste compost as the residue medium for *Pleurotus eryngii* planting, which was used as a feed replacement; its consequent influence on broiler chickens’ intestinal microbiota, anti-inflammatory responses, and anti-oxidative status was likewise studied. A total of 240 male broilers were used and allocated to four treatment groups: the basal diet—control group (corn–soybean); 5% replacement of a soybean meal via PWMC (*Pennisetum purpureum* Schum No. 2 waste mushroom compost); 5% replacement of a soybean meal via FPW (*Saccharomyces cerevisiae* fermented PWMC); 5% replacement of a soybean meal via PP (*Pennisetum purpureum* Schum No. 2). Each treatment had three replicates and 20 birds per pen. The levels of glutathione peroxidase and superoxide dismutase mRNA as well as protein increased in the liver and serum in chickens, respectively; mRNA levels of inflammation-related genes were also suppressed 2 to 10 times in all treatments as compared to those in the control group. The tight junction and mucin were enhanced 2 to 10 times in all treatment groups as compared to those in the control, especially in the PWMC group. Nevertheless, the appetite-related mRNA levels were increased in the PWMC and FPW groups by at least two times. In ileum and cecum, the Firmicutes/Bacteroidetes ratios in broilers were decreased in the PWMC, FPW, and PP groups. The *Lactobacillaceae* in the ileum were increased mainly in the PWMC and control groups. Overall, high-fiber feeds (PWMC, FPW, and PP) could enhance the broilers’ health by improving their antioxidant capacities and decreasing their inflammatory response as compared to the control. Based on the results, a 5% replacement of the soybean meal via PWMC is recommended in the broiler chickens’ diet.

## 1. Introduction

Agricultural by-products such as wheat bran, crop hull, straw, etc., are inevitable substances in the agricultural process, but they are often not effectively used [1,2]. On the other hand, feed accounts for at least half of costs in the production of animal-related products [1,2]. However, with the increasing awareness of environmental protection, agricultural wastes or by-products are required to be properly treated [2,3,4]. In order to effectively reduce animal production costs and increase the value of agricultural by-products, previous studies have suggested the use of by-products such as mushroom waste compost since it is rich in mycelium and contains abundant functional secondary metabolites, which can improve antioxidant capacities, and regulate inflammatory response as much as animal feed ingredients [3,4,5].

Carbohydrates, one of the main sources of energy for animals, can be roughly divided into two broad categories. The first category is the source of energy for animals such as starch, glucose, and sucrose, which are decomposed by enzymes produced by the animals. The second category is the dietary fiber that cannot be digested by the animal itself [6]. A distinction can also be made between hard-to-digest fibers such as cellulose, chitin, and other fibers, depending on whether the microorganism can be digested [5,6]. However, high-fiber agricultural by-products may reduce the performance of monogastric animals in traditional concepts [6], so this proposal has not been favored by animal producers for a long time. Nevertheless, there are increasingly more studies that have used agricultural by-products as the source of animal feed ingredients, and results have shown no reduction in the production performance of animals and that these could even help improve animal health [7,8]. It has been suggested that high-fiber feed ingredients could help to strengthen the intestinal barrier and antioxidant capacity of animals, and also reduce inflammation [4]. The additional antioxidant capacity can help in conditions of sudden oxidative stress, whether the negative factors come from the environment or pathogenic bacteria [9]. High-fiber feed ingredients also help with muscle formation and reduce adipose accumulation, thereby improving animal body composition [10]. In addition, the use of probiotics or enzymes to initially degrade agricultural by-products in vitro further increases the utilization of agricultural by-products [11].

When animals are subjected to environmental stress such as pathogen infection or heat stress, stimulation causes an increase in oxidative stress and inflammation [12,13]. When stimulated by pathogenic bacteria, the animal initiates an immune response, which leads to a cytokine storm. However, excessive inflammation can reduce animal performance and even lead to death [14,15]. On the other hand, when the oxidative pressure is high, animals are not able to eliminate the damage caused by free radicals to cells and/or organs [16]. Moreover, the antioxidant system is mainly regulated by the liver; thus, the antioxidant capacity is also related to liver performance in animals [16,17,18]. Fiber could reduce the inflammatory response and increase the antioxidant capacity of the serum and the liver, as well as enhance the growth of intestinal villi and the overall health of animals [11]. Therefore, in the recent decade, more attention has been paid to the influence of gut microbiota composition on animal health [19]. Previous studies have pointed out that gut microbiota composition is not only related to animal health, anti-inflammatory levels, and antioxidant capacity, but also affected animal appetite, the circadian clock, and mood [19]. The intestinal microbial composition can be changed rapidly and directionally by altering dietary patterns [7]. Under the combined effects of microorganisms and diet, the metabolome of animals may be changed [20]. Studies of mammals have shown that the increased concentration of dietary fiber can promote the growth of Bacteroides, which reduce the number of Firmicutes in the gut [21]. Bacteroides are related to animals’ body maintenance, while Firmicutes are positively related to the degree of animal obesity [21]. In addition, increasing the dietary fiber concentration of animals also helps the growth of *Bifidobacterium*, which has a positive effect on animal health [21].

Although considerable research has been conducted on the effects of gut microbiota composition on mammalian health, only a few studies have been performed on poultry. Therefore, the purpose of this study was to investigate the effects of high-fiber agricultural by-products on the broilers’ microbial activity under different treatments and to evaluate its anti-inflammatory and anti-oxidative effects in broilers.

## 2. Materials and Methods

The entire protocol for animal feeding was in accordance with rules established by the Animal Care and Use Committee at the NCHU (IACUC: 108−049). In order to evaluate the effects of PWMC (*PP waste mushroom compost*), FPW (*Saccharomyces cerevisiae fermented PWMC*), and PP (*Pennisetum purpureum Schum No.2*) on broilers’ antioxidant and anti-inflammatory capacities, intestinal tight junction expression, gut microbiota, and morphology were measured at the National Chung Hsing University (NCHU), Taiwan.

### 2.1. Collection and Characteristics of Pennisetum Purpureum Schum No.2 (PP), PP Waste Mushroom Compost (PWMC), and Saccharomyces Cerevisiae Fermented PWMC (FPW)

The PWMC is a kind of waste medium, which contains at least 70% PP after *Pleurotus eryngii* planting. Mature *Pleurotus eryngii* was removed and the medium was collected, dried, and stored at 4 °C in a refrigerator before use. The PWMC came from the Taiwan Agricultural Research Institute Council of Agriculture, Executive Yuan.

The production of FPW was accomplished according to the method described by Chuang et al. [11], which was slightly modified for this study. PWMC was collected by the methods mentioned above; 1 mL of 10^8^ CFU/mL *S. cerevisiae* was added to 100 g PWMC, and the moisture was adjusted to 60%. After culturing at 30 °C for 3 days, the fermented product was dried in an oven at 50 °C for 1 day and stored at −20 °C in a refrigerator before use. Before the test, we confirmed that the FPW contained at least 10^8^ CFU/g of *S. cerevisiae.*

The chemical composition (dry matter (DM), crude protein (CP), ether extract (EE), crush ash, neutral detergent fiber (NDF), and acid detergent fiber (ADF)) as well as the functional compounds (crude polysaccharide and volatile fatty acids) in PWMC, FPW, and PP were analyzed according to the methods described in AOAC (Assoc. Offic. Anal. Chem) [22] or the methods described below.

#### 2.1.1. Crude Polysaccharide Measurement

The methods of pentose (D(+)-Xylose, Wako, Osaka, Japan) and hexose (D-(+)-Glucose, Sigma-Aldrich, St. Louis, MO, USA) measurement were modified based on the phenol-sulfuric acid method of Nielsen [23]. Briefly, the sample was extracted by deionized water at 95 °C for 30 min and cooled down to room temperature before use. After that, 1 mL 5% phenol solution and 5 mL sulfuric acid were added to PWMC, FPW, and PP water extracts. After incubation for 15 min, the absorbance was measured at 480 nm with U-2900 Spectrophotometer (Hitachi, Tokyo, Japan) for the standard curveof xylose and glucose

#### 2.1.2. Phenolic Components

The PWMC, FPW, and FF water extracts (extracted at 95 °C for 30 min with deionized water) were filtrated by a 0.22 μm filter before use for the detection of catechins. The phenolic component content in the water extracts of PWMC, FPW, and FF were detected by HPLC (High Performance Liquid Chromatography) (Hitachi, Tokyo, Japan) with a pump (5110), a column (C18-AR, 250 × 4.8 mm, maintained at 40 °C by the column oven (5310)), an autosampler L-2200 (Hitachi, Tokyo, Japan), and a computer system with HPLC D-2000 Elite (Hitachi, Tokyo, Japan). The conditions of the mobile phase were (A) 0.05% *v/v* H3PO4 and (B) 3:2 *v/v* CH3OH/CH3CN solution, 1.0 mL/min, and UV detection at 280 nm. The known concentration (0.01–1.5 mg/mL) of gallocatechin (GC), epigallocatechin (EGC), catechin (CC), epicatechin (EC), gallic acid (GA), epigallocatechin gallate (EGCG), epicatechin gallate (ECG), catechin gallate (CG), and caffeic acid were also measured by the methods described above for the standard curve.

#### 2.1.3. Free Radical Scavenging Ability of 2,2-. Diphenyl-1-Picrylhydrazyl (DPPH)

The assay for DPPH (2,2-diphenyl-1-picrylhydrazyl) scavenging capacity was conducted and slightly modified according to a previous report [24]. To deionize water, 1 g sample was added and soaked at 95 °C for 1 h in Tempette Junior TE-8J Heating Water Bath (Techne, Staffordshire, UK). After cooling, it was centrifuged at 3000 rpm× *g* for 10 min with Heraeus Megafuge 8 Centrifuge (Thermo Fisher Scientific Inc., Waltham, MA, USA). The supernatant was added to 0.1 mM DPPH and after incubation for 30 min, the absorbance was measured at 517 nm with a U-2900 Spectrophotometer (Hitachi, Tokyo, Japan). Butylated hydroxytoluene (BHT) was taken as the positive control. DPPH scavenging capacity (%) = (1 − sample 517 nm OD/control 517 nm OD) × 100%).

#### 2.1.4. Reducing Powder

The measurement of the reducing powder was performed as described by the Oyaizu [25]. Briefly, the sample was extracted in deionized water at 95 °C for 30 min and cooled down to room temperature before use. Four milliliters of sample solution, 1 mL 0.2 M PBS (pH 6.6), and 1 mL 1% potassium ferricyanide were mixed and incubated at 50 °C for 20 min in Tempette Junior TE-8J Heating Water Bath (Techne, Staffordshire, UK). The incubated mixture was cooled down with ice and 1 mL trichloroacetic acid was added. The new mixture was centrifuged at 3000 rpm× *g* for 10 min with Heraeus Megafuge 8 Centrifuge (Thermo Fisher Scientific Inc., Waltham, MA, USA) and 5 mL supernatant was moved to a new tube. The supernatant was mixed with 1 mL 0.1% ferric chloride and kept at room temperature for 10 min. The absorbance of the sample at 700 nm was measured with U-2900 Spectrophotometer (Hitachi, Tokyo, Japan) and compared to the results of BHT.

#### 2.1.5. Malondialdehyde Production

The method used for malondialdehyde (MDA) measurement was based on the use of thiobarbituric acid reactive substances (TBARS) described by Zeb and Ullah [26]. Briefly, the sample was extracted in deionized water at 95 °C for 30 min and cooled down to room temperature before use. One milliliter of the sample solution was mixed with 4 mL 1% lecithin, 20 mM Na_2_HPO_4_-NaH_2_PO_4_ buffer, 0.2 mL 25 mM FeCl_3_, and 2.5 mL 20 mM Na_2_HPO_4_-NaH_2_PO_4_ buffer (pH 7.4). The mixture was incubated at 37 °C for 2 h in Tempette Junior TE-8J Heating Water Bath (Techne, Staffordshire, UK) and mixed with 2 mL 20 mg/mL BHT, 4 mL 1% thiobarbituric acid, and 2 mL 2.8% TCA to stop the reaction. The absorbance of the mixture was measured at 532 nm with U-2900 Spectrophotometer (Hitachi, Tokyo, Japan) and the rate of inhibition of MDA production was calculated.

#### 2.1.6. Ferrous Chelating Capacity

The ferrous chelating capacity assay was performed according to the methods described by Yuris and Siow [27]. Briefly, the sample was extracted in deionized water at 95 °C for 30 min and cooled down to room temperature before use. One milliliter of the sample solution was mixed with 0.1 2 mM ferrous chloride solution and 3 mL deionized water. The mixture was kept at room temperature and protected from light for 10 min. Then, 0.2 mL ferrozine was added to the mixture and the absorbance was measured at 562 nm with U-2900 Spectrophotometer (Hitachi, Tokyo, Japan). The following formula was used to calculate the ferrous chelating capacity:Ferrous chelating capacity (%) = [(A0 − A1)/A0]/100(1)
where A0 is the absorbance of the positive control (Ethylenediaminetetraacetic acid), and A1 is the sample absorbance.

### 2.2. Animal Experiment Design

The animal experiment design was slightly modified and followed by the procedure described by Chuang et al. [11]. Briefly, a total of 240 male broilers (Ross 308) were used and allocated to 4 different treatments: the corn–soybean basal diet, 5% replacement of a soybean meal via PWMC, FPW, and PP. Each treatment had three replicates (20 broilers for each pens). The initial weight of 1-day-old chicks for each treatment was similar (48.0 ± 0.7 g/bird). The room temperature was controlled throughout the experiment (33 °C for 1-day-old chicken, and decreased slowly to 22 °C after 30 days). Each group had similar crude protein and gross energy content in both starter (from 1 to 21-day-old) and finisher (from 22 to 35-day-old) feed; furthermore, the nutrient concentration of the broilers’ diet was achieved or exceeded nutrient requirement according to NRC 1994 (Table 1). Feed for the different treatments was recalculated and the approximate composition (energy equivalent to 3050 vs. 3175 kcal/kg DM and crude protein at 23 vs. 21% DM for the starter and finisher periods, respectively) were analyzed according to the methods described in AOAC [22] and shown in Table 1. The serum, chyme in ileum and cecum, liver, spleen, hypothalamus, ileum, and jejunum were collected from 35-day-old broilers for the follow-up analysis.

### 2.3. Serum Characteristics

Serum from broilers was collected from a total of 36 chickens (three from each pen, nine for each treatment) following the methods described by Chuang et al. [11]. Briefly, chicken blood was collected and stored at 4 °C in a refrigerator R2551HS (TECO Electric and Machinery Co., Ltd., Taipei, Taiwan) for 4–5 h. Blood samples were centrifuged at 3000 rpm× *g* for 10 min in order to separate the blood cells and the serum, and the serum was stored at −20 °C in a refrigerator before being analyzed. The concentration of tumor necrosis factor-alpha (TNF-α), glutathione peroxidase (Gpx), and superoxidase dismutase (SOD) in broiler serum were measured (TNF-α, Gpx, and SOD were from Cayman Chemical Co., Ltd. Ann Arbor, MI, USA). All the methods for analyzing these compounds followed the manufacturer’s protocol. Other serum characteristics were measured with an automatic biochemical analyzer (Hitachi, 7150 auto-analyzer, Tokyo, Japan).

### 2.4. DNA Extraction and 16S rRNA Gene Sequencing

Ileal and cecal digesta in 35-day-old broilers were collected from 6 broilers in each of the four (control, PWMC, FPW, and PP) groups. Samples were immediately isolated for their genomic DNA by using Quick-DNA Fecal/Soil Microbe Miniprep Kits (Zymo, Irvine, CA, USA); procedures were accomplished by following manufacturer’s protocol. Extracted DNA qualities, including purity and concentration, were analyzed through the NanoDrop 2000 spectrophotometer (Thermo Scientific, Waltham, MA, USA). Briefly, ileum or cecum chyme from 35-day-old broilers was collected in collection tubes provided by the manufacturer and stored at −80 °C in a refrigerator before being analyzed. After a series of dissolution, filtration, and extractions, pure microbial DNA from broiler ileum or cecum chyme was isolated. The concentration and purity of purified DNA were measured with the absorbance ratio of 260/280 nm. For the identification of microbial species, the V3–V4 hypervariable regions of 16S rRNA were sequenced. The Qubit^®^ 2.0 Fluorometer (Thermo Scientific) and an Agilent Bioanalyzer 2100 system were used for the library quality and the final library was sequenced by the IlluminaHiSeq2500 platform with 250 bp paired-end reads. After sequencing, the UCHIME algorithm was used to detect chimera sequences that were then removed to obtain effective tags. Uparse (Uparse v7.0.1090; http://drive5.com/usearch/, accessed on 11 June 2021) software was used for the sequence analysis. Microbes with operational taxonomic units (OTUs) at 97% similarity were clustered using USEARCH (version 7.0.1090). The representative sequence of the OTUs was selected for further annotation. The difference of alpha-diversity (Shannon index, Observed species, Simpson, and Abundance-based coverage estimator (ACE)) were analyzed by QIIME (Version 1.9.1) and displayed by R software (v.3.3.1) using the alpha function in the microbiome R package. The relative abundance of the OTU in every treatment was evaluated by metagenomeSeq.

### 2.5. Total RNA Isolation, qPCR, and Sequencing

The mRNA of 35-day-old broilers was collected from 6 broilers from each of the four groups and isolated from the liver (for antioxidant capacities measurement), spleen (for anti-inflammatory capacities measurement), ileum (for tight junction expression measurement), and hypothalamus (for appetite regulation measurement), respectively. Among these, the mRNA collected from the hypothalamus was removed from broilers before (H1) and after (H2) fasting for 24 h. During fasting, the birds were provided with water spray and ventilation. The methods of mRNA isolation and qPCR measurement were both described by Chuang et al. [11]. Briefly, total mRNA from each tissue (approximately 100 mg for individual tissues) was isolated following the manufacturer’s protocol (SuperScript™ FirstStrand Synthesis System reagent, Invitrogen, Carlsbad, CA, USA) and purity was determined with the absorbance ratio of 260/280 nm with Epoch Microplate Spectrophotometer (BioTek Instruments, Winooski, VT, USA). The PCR conditions were as follows: initial denaturation at 95 °C for 20 s, 40 cycles of 95 °C for 3 s, 55 °C for 30 s, and 72 °C for 30 s, and then final extension at 72 °C for 5 min using StepOnePlus™ Real-Time PCR System (Thermo Fisher Scientific Inc., Waltham, Massachusetts, USA). After synthesizing cDNA from mRNA, cDNA was mixed with 2× SYBR GREEN PCR Master Mix-ROX (Gunster Biotech, Co., LTD, New Taipei City, TW), deionized water, and each primer in the ratio of 25:6:9:5. StepOnePlus™ Real-Time PCR System (Thermo Fisher Scientific Inc., Waltham, MA, USA) was used to assess the qRT-PCR performance. After these, 2-^ΔΔCt^ were measured in each group, and the relative expression level of each mRNA was calculated and compared to that of the control group. The methodology was as described by Schmittgen and Livak [28]. ß-actin was used as a housekeeping gene for normalization. All primes were designed according to the genes of *Gallus* (chicken) from GenBank (Table 2). After sequencing, whole tags were assembled using the UCHIME algorithm to detect chimera sequences; the chimera sequences were removed before the effective tags were obtained. Sequence analysis was performed using the Uparse software (Uparse 135 v7.0.1090; Available online: http://drive5.com/usearch/ (accessed on 15 January 2020)). Sequences with ≥ 97% similarity were assigned to the same operational taxonomic units (OTUs). A representative sequence of each OTU was selected for further annotation. Alpha diversity was applied to analyze the complexity of the species diversity of a sample by using six indices: Observed OTUs, Shannon, Simpson, Chao1, ACE, and PD whole tree. All the indices of our samples were calculated using Quantitative Insights Into Microbial Ecology (QIIME, v1.9.1). To evaluate differences in samples with respect to species complexity, beta diversity analysis on weighted unifrac was conducted using the QIIME software (v1.9.1). A principal coordinates analysis (PCoA), using the R software (v.3.3.1) with the ade4 and ggplots packages, was performed at the genus level. LEfSe (Linear discriminant analysis effect size) was performed to detect differential abundant taxa across groups using the default parameters.

### 2.6. Statistical Analysis

The data collected were statistically analyzed using the general linear models procedure of the SAS software (SAS^®^ 9.4, 2018) following a Completely Randomized Design (CRD). Variables of serum, intestine, and gut microbiota were tested for normal distribution and homogeneity. Data on the dietary treatments were subjected to analysis of variance using the Statistical Analysis System Institute Package (SAS) and the mean values were compared using Duncan’s range test, with the significant level at *p* < 0.05.

The mathematic model was as follows:Yij = µ + Di + εij
where Yij is the average of birds in pen j and their dietary treatment i; µ is the overall mean; Di is the fixed effect of the dietary treatment (1: control, 2: PWMC, 3: FPW, and 4: PP); εij is residual error when the pen was regarded as an experimental unit, (0, σ2ε).

## 3. Results

### 3.1. The Characteristics of PWMC, FPW, and PP

There were several functional components in PWMC, FPW, and PP (Table 3). All PWMC, FPW, and PP had a high content of polysaccharide (85, 83, and 88 glucose mg/g DM, respectively), total phenol content (1.8, 1.8, and 2.1 GAE mg/g DM, respectively, data not shown), and total flavonoids (1.2, 1.3, and 1.7 QE mg/g DM, respectively, data not shown). The phenol content in PWMC, FPW, and PP, including GC, EGC, CC, EC, GA, EGCG, ECG, CG, and caffeic acid, were estimated. Among them, there was at least 0.1% EGC and GC content in PWMC, FPW, and PP, and about 0.02% EC and EGCG in PWMC, FPW, and PP. Particularly, after being fermented by *S. cerevisiae*, the phenol-like chemical increased by about 1.2 to 2 times in PWMC. Some natural organic acids were also present in PWMC, FPW, and PP. However, after fermentation by *P**leurotus*
*eryngii* or *S. cerevisiae*, the volatile fatty acids increased much more in PWMC and FPW than in PP (31, 28, and 12 μM/g, respectively). Fermentation by *S. cerevisiae* could also increase the hemicellulose content from about 11% to 20% in FPW as compared to that in PWMC (data not shown, calculated by the difference between NDF and ADF).

Both phenols and flavonoids were found to have high antioxidant capacities. The antioxidant capacities are shown in Figure 1, including the ferrous ion chelation rate (Figure 1A), Di (phenyl)-(2,4,6-trinitrophenyl) iminoazanium (DPPH) scavenging capacities (Figure 1B), total reducing capacities under 700 nm absorbance (Figure 1C), and malondialdehyde production inhibition rate (Figure 1D). Among them, PWMC, FPW, and PP had similar and high antioxidant capacities in all experiments.

### 3.2. Serum Characteristic of Broilers

The data of the serum characteristics of 35-day-old broilers showed that an increase in the fiber content of broiler feed could significantly decrease the glucose concentration in the broilers’ blood as compared to that in the control group (from 276 mg/dL to 236, 240, and 227 mg/dL; *p* < 0.05). Nevertheless, the triglyceride concentration in the broilers’ blood also decreased by about 25%, in both the control group and in the other treatments (55, 41, 38, and 40 mg/dL; *p* < 0.001). As the classic antioxidant enzyme in animal, the concentration of SOD in the broilers’ serum was increased in the FPW and PP groups (*p* < 0.001), but not in the PWMC group, as compared to that in the control group. The concentration of TNF-α in the serum of the broilers also decreased significantly (*p* < 0.05) compared to that in the control group (Table 4); however, the interleukin 1 beta (IL-1β) was underdetermined in each treatment (not shown in data).

### 3.3. Gut Microbiota Composition and Function

#### 3.3.1. Microbiota Composition in the Ileum of Broilers

According to Figure 2A,B, comparing with other groups, the FPW group has the highest (*p* < 0.05) relative abundance of species in the broilers’ ileum, followed by the PP and PWMC groups. Among them, only three different microbes were detected in the control group, while FPW had 70 species (Figure 2B). The Firmicutes/Bacteroidetes ratios decreased (*p* < 0.05) in the PWMC, FPW, and PP groups, but the FPW group showed a higher (*p* < 0.05) standard error range (Figure 2C). At the phylum level, the relative abundance of Proteobacteria seemed hardly changed; however, the *Epsilonbacteria* and *Bacteroidetes* increased (*p* < 0.05) in PWMC, FPW, and PP. Among them, the relative abundance of Cyanobacteria increased (*p* < 0.05) only in the PP group (Figure 2D). At the genus level, the abundance of *Romboutsia* seemed similar to that of the control, FPW, and PP groups, but decreased (*p* < 0.05) in the PWMC group. The relative abundance of *Campylobacter* increased in broilers after a high-fiber treatment (PWMC, FPW, and PP), especially in the FPW group. Among them, the *Eimeria praecox* increased only in the PP group. The relative abundance of *Candidatus arthromitus* was 10 times higher (*p* < 0.05) in PWMC, and especially in the FPW and PP groups, as compared to that in the control group. Furthermore, *Turicibacter* decreased (*p* < 0.05) mostly in PP, followed by FPW and PWMC, as compared to that in the control group (Figure 2E,H). Particularly, at the order and family levels, the relative abundance of *Lactobacillales* and *Lactobacillaceae*, respectively, increased (*p* < 0.05) in the control and PWMC groups (Figure 2F,G). According to the alpha diversities shown in Figure 2I–L, the PWMC, FPW, and PP groups showen higher (*p* < 0.05) amounts of observed species and ACE index compared to those of the control group; however, the difference was not significant. Each group had a similar score in the Shannon and Simpson indices.

#### 3.3.2. Microbiota Composition in the Cecum of Broilers

The microbial relative abundance in the cecum was similar in each group (Figure 3A,B), which was in contrast to the data shown in the ileum (Figure 2A). However, the Firmicutes/Bacteroidetes ratio still decreased (*p* < 0.05) in the high-fiber groups (Figure 3C). According to the relative abundance shown in Figure 3D, we found that the percentage of Bacteroidetes and *Verrucomicrobia* increased (*p* < 0.05). At the genus level, almost 50% of the microbes occurred in all groups, in the control as well as in the other treatments. The relative abundance of *Akkermansia* increased (*p* < 0.05) in the high-fiber treatments, especially the FPW. Other microbes that increased (*p* < 0.05) in relative abundance included Ruminococcaceae_UCG_014, *Lactobacillus*, and *Barnesiella*. However, the *Faecalibacterium* decreased (*p* < 0.05) only slightly in the high-fiber treatments as compared to the control group (Figure 3E). According to the alpha diversities shown in Figure 3I–L, the treatment groups still had the higher number of observed species and ACE index (*p* < 0.05), but the standard error in the FPW group was much higher than in the other groups (*p* < 0.05) (Figure 3I,J).

### 3.4. mRNA Expression in Liver, Spleen, and Ileum of Broilers

Concerning mRNA expression in the spleen of 35-day-old broilers, the mRNAs of most of the inflammation-related proteins decreased significantly (*p* < 0.05). Among them, the IL-1β, inducible nitric oxide synthases (iNOS), and nuclear factor kappa B (NFκB) decreased (*p* < 0.05) by over 10 times after each treatment as compared to those in the control group (iNOS mRNA expression did not differ significantly in only PWMC and the control). Furthermore, the level of interferon-gamma (IFN-γ) mRNA decreased (*p* < 0.05) only in the FPW group (Figure 4).

The nuclear factor erythroid 2-related factor 2 mRNA expression in the broilers’ liver increased 4–5-fold in the high-fiber treatment groups as compared to the control group; therefore, the glutamate-cysteine ligase catalytic (GCLC) and heme oxygenase-1 (HO-1) mRNA expressions also increased. Among them, GCLC increased significantly only in the FPW group, and HO-1 increased the most in the PP group. Furthermore, the GPx and SOD mRNA expressions in the broilers’ liver also increased significantly (*p* = 0.003 and < 0.001). High-fiber feed replacement could significantly increase the tight junction and mucus, including claudin 1, mucin 2 (MUC2), as well as occludin and zonula occludens 1 (ZO-1) mRNA expressions in broilers (*p* < 0.001, < 0.01, < 0.001, and < 0.005, respectively) (Figure 4).

Among the expressions of appetite regulation-related mRNAs, there was no significant difference in the leptin (LEP) expression in the hypothalamus of the broilers before fasting; however, after fasting for 24 h, the LEP decreased by over 2-fold in the PWMC and FPW groups as compared to the control group, but not in the PP group (*p* < 0.001). Furthermore, the neuropeptide Y (NPY) expressions in PWMC and FPW were higher (*p* < 0.05) than in the control group, both before and after fasting; however, in the PP group, NPY increased only after fasting for 24 h. The pro-opiomelanocortin (POMC) and cocaine- and amphetamine-regulated transcript (CART) mRNA expressions in all treatments were lower (*p* < 0.05) than those in the control group. However, among the PWMC, FPW and PP treatment, the expression of POMC and CART mRNA in PP was still higher (*p* < 0.05) than in PWMC and FPW groups both before and after fasting (Figure 4).

## 4. Discussion

There are plenty of edible mushrooms worldwide; however, only a few mushrooms are suitable for planting and can be widely accepted by consumers [4]. Among them, *Pleurotus eryngii* is one of the most popular edible mushrooms [4]. Previous studies have shown that *Pleurotus eryngii* contains functional compounds such as triterpenes and phenol, which could increase the antioxidant capacities of animals and improve animal health [28]. When planting mushrooms, the cost and medium composition should be considered [2]. As a cheap, high-fiber content and several functional compounds, *Pennisetum purpureum* Schum No.2 was considered a good substance in which to culture mushrooms [2]. PWMC was the medium in which *Pleurotus eryngii* had been planted; thus, it contained a high amount of myelin in several functional compounds (Table 3). However, in order to maintain the growth of *Pleurotus eryngii*, the residue medium must contain higher cellulose and lignin (Table 3). This situation would be ameliorated by *S. cerevisiae* fermentation. FPW showed higher hemicellulose content, and more extractable EGCG-like compounds that were released from PWMC. Earlier studies showed that the concentration of extractable functional compounds (phenol and phosphorus, especially) would increase on fermented prebiotics [29]. Phenol and flavonoids have been shown to enhance antioxidant capacities both in vitro and in vivo [30]. By chelating ions, phenol could decrease the lipid oxidation rate, thereby decreasing the MDA content [9,31]. Furthermore, PWMC, FPW, and PP were also good at decreasing the free radicals from the DPPH. These results show that PP contains phenol-like compounds, which has slightly increased level after culturing with *Pleurotus eryngii* and *S. cerevisiae* fermentation.

In Table 4 and Figure 4, we observed that the antioxidant-related enzyme and mRNA levels were increased, and a decrease in inflammation. In animals, the antioxidant system can be divided into two categories: enzyme and non-enzyme systems [9]. Among the enzymes, catalase could decrease the accumulation of hydrogen peroxide; the non-enzymatic category includes glutathione (GSH), which is well-known to reduce free radicals. Both of these help in reducing reactive oxygen species (ROS) production and in enhancing animal health [24]. Nrf2 is a common transcription factor upstream mRNA that can enhance the production of a series of antioxidant proteins such as GCLC and HO-1 by binding to the antioxidant responsive element (ARE) in the nucleus [32]. HO-1 could decrease the oxidant stress in animals, and the amount of GCLC is positively correlated with GSH content. GSH could bind to electrons on ROS and become GSSH to reduce the damage due to ROS. Gpx is the key enzyme that can oxidize GSSH in order to form GSH [31]. Although the Gpx protein increased only slightly in the broilers’ serum, the Gpx mRNA in the broilers’ liver increased significantly in the high-fiber treatment groups. Furthermore, both SOD protein and mRNA increased in broilers on the high-fiber treatment, especially in the FPW and PP groups, which could protect the broilers from ROS by transferring O- to H_2_O_2_ and further provide the substrate for the catalase [31]. In addition, inflammation is a double-edged sword—it does not only help animals to face a pathogen but it may also cause further cell damage [9]. Based on this, it can thus be understood that when an animal finds itself in a normal situation, oversensitive inflammatory reactions are not desired [9]. Therefore, under normal conditions, a decrease in inflammation levels in animals should be considered. PWMC, FPW, and PP could decrease both the protein and mRNA levels of TNF expression. Furthermore, the main upstream inflammatory signals of mRNAs, IL-1β, and NFκB all decreased significantly under a high-fiber treatment, and further decreased the iNOS mRNA expression [9]. It is worth noting that IFN-γ only decreased in the FPW group. This could be due to the glucan and mannan content in *S. cerevisiae*, which can help to encase lipopolysaccharides in the environment. Similar results were shown by Chuang et al. [11]; the addition of probiotic (10^6^ CFU/kg feed of *S. cerevisiae*) fermentation could decrease the IFN-γ by over 10 times. Similar results in the broilers’ gut barrier function were shown; among them, FPW had the highest claudin-1 mRNA expression. Enhanced gut barrier function can help in animal defense against infection from pathogens in the gut and decrease the level of inflammation [7]. With no inflammation, the broilers’ feed intake increased. POMC is one of the main appetite regulation peptides in animals and is negatively correlated with animal feed intake by inducing CART and MSH family production [33,34]. Furthermore, LEP is well-known for its anti-obese function in mammals and was considered to be present in poultry until 2014 [35]. By improving satiety and accelerating muscle formation in animals, LEP could effectively decrease animal body weight. In contrast to these, NPY is the main neuropeptide that can enhance animal feed intake [33]. Our results showed that before fasting, the POMC and CART mRNA expressions decreased and that of NPY increased mainly in the PWMC and FPW groups. However, when fasting for 24 h, the LEP further decreased in the PWMC and FPW groups, and the NPY mRNA expression also increased in the PP group. These data reveal that PWMC and FPW could enhance appetite-inducing mRNA expression, while PP appears to maintain normal appetite levels in broilers. The possible reason may be the regulation of animal appetite not only through endocrines but also at the nutrient level and other physiological levels [36].

Compared to the ileum, the cecum is the main site for fermentation and contains a large number of microbes [37]. Therefore, it seems that the high-fiber feed mainly affects the relative abundance of microbes in the ileum (Figure 2 and Figure 3). As expected, the Bacteroidetes increased in the high-fiber groups as compared to the control group in both ileum and cecum. Previous studies have shown that the dietary fiber in food is positively correlated with the abundance of Bacteroidetes, and increasing the ratio of Bacteroidetes/Firmicutes further enhances animal health and downregulates the inflammatory response [7,21]. Furthermore, the abundance of Bacteroidetes is also correlated with lower body weight [21], which probably because of an increase in the rate of lipolysis. In fact, the accumulation of subcutaneous fat in PWMC, FPW, and PP decreased by about 30%, and the percentage of the inner breast in the broilers’ carcass increased slightly (not shown in data). *Turicibacter* is a common bacterium in the animal gut, which is positively related to colitis [38]. However, the number of *Turicibacter* in the ileum of broilers decreased in the high-fiber groups, which could have consequently decreased the inflammatory level in broilers. Desai et al. [7] also indicated that an increase dietary fiber could decrease the level of colitis.

*Lactobacillus* increased in relative abundance in the ileum of the control and PWMC groups, and in the cecum of the PWMC and PP groups. The possible reason is that *S. cerevisiae* competes with *Lactobacillus* because both of them are fiber- prefer microbes [39,40]. Furthermore, PWMC could provide more growing substrates for *Lactobacillus* than PP could, thus explaining the growth of *Lactobacillus* in the ileum of the PWMC group. Interestingly, accelerated lipolysis, decreased inflammation levels, and the increase in the number of *Lactobacillus* could be directed to promote animal health and effectively reduce fat accumulation. Overall, it seems that high-fiber feed could enhance animal antioxidant capacities, decrease inflammatory levels, and alter gut microbes both in the ileum and the cecum, thus leading to improv animal health. However, the types and sources of fibers also limit the availability for products. For example the fibers with too high a molecular weight of polymerization may be difficult digested and utilized in a short time of animals in diet. Moreover, many soluble non-starch polysaccharides (NSP) may lead to excessive fermentation by gut microorganisms and reduce growth/production performances in animals [15,21]. Therefore, in the study when considering the commercial aspect, PWMC—on which *Pleurotus eryngii* was first cultured—did not need further fermentation, and yielded similar antioxidant capacities and slightly higher growth performance in 35-day-old broilers. Therefore, a 5% replacement of a soybean meal via PWMC is recommended in broilers’ feed as opposed to other groups.

## 5. Conclusions

Our results show that high-fiber feeds (PWMC, FPW, and PP) could enhance animal health by inducing antioxidant capacities and decreasing the inflammatory response in broilers as compared to the control group. Furthermore, the number of *Lactobacillus* (ileum) increases in the FPW and PP groups, also increases (cecum) in the PWMC and FPW groups. Additionally, this study showed that in order to enhance the broilers’ health, a 5% replacement of a soybean meal via PWMC is recommended for the broilers’ diet.

## Figures and Tables

**Figure 1 animals-11-02550-f001:**
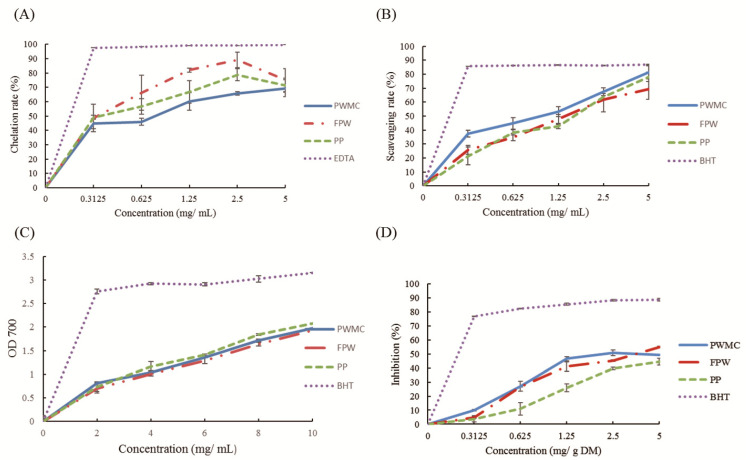
Antioxidant capacity analysis in *Pennisetum purpureum* Schum No.2 waste mushroom compose (PWMC), *Saccharomyces cerevisiae* fermented PWMC (FPW), and *Pennisetum purpureum* Schum No.2 (PP). (**A**) Ferrous ion chelation rate; (**B**) Di (phenyl)-(2,4,6-trinitrophenyl) iminoazanium (DPPH) scavenging capacities; (**C**) Total reducing capacities under 700 nanometer absorbance; (**D**) Malondialdehyde production inhibition rate measured by 2-thiobarbituric acid reacting substances test (TBARS). Results are means of five samples from each analysis. *n* = 3.

**Figure 2 animals-11-02550-f002:**
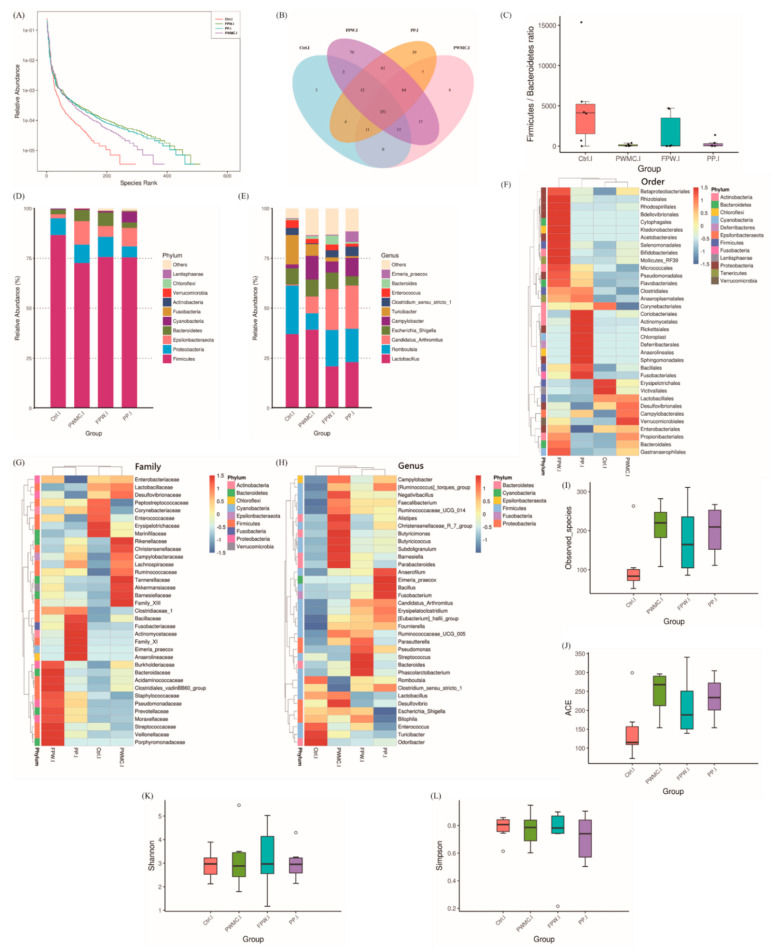
The 35–day–old broilers’ ileum microbiota influenced by a corn–soybean basal diet (Ctrl), *Pennisetum purpureum* Schum No. 2 waste mushroom compost (PWMC), *Saccharomyces cerevisiae* fermented PWMC (FPW), and *Pennisetum pur--pureum* Schum No. 2 (PP). (**A**) Microbe relative abundance; (**B**) Microbe Venn diagram; (**C**) Firmicutes to Bacteroidetes ratio; (**D**,**E**) Relative abundance of each microbe in the broilers’ ileum at the phylum (**D**) or genus (**E**) level; (**F**–**H**) Taxa heatmap of ileum microbes according to order (**F**), family (**G**), and genus (**H**). (**I**–**L**) Alpha diversity of 35–day–old broil–ers’ ileum microbes in each treatment; (**I**) Observed species, (**J**) Abundance-based coverage estimator (ACE), (**K**) Shannon index, and (**L**) Simpson index. Results are means of six samples obtained from the individual birds from the control and experimental groups (*p* < 0.05).

**Figure 3 animals-11-02550-f003:**
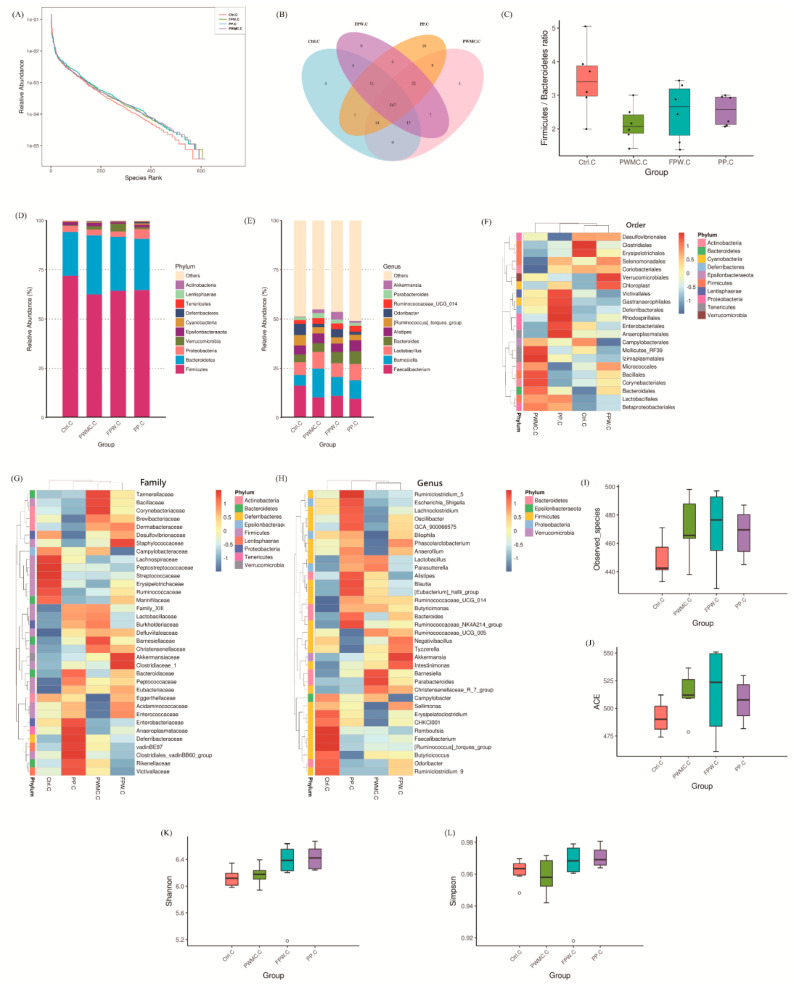
The 35–day–old broilers’ cecum microbiota influenced by a corn–soybean basal diet (Ctrl), *Pennisetum pur*–*pureum* Schum No. 2 waste mushroom compost (PWMC), *Saccharomyces cerevisiae* fermented PWMC (FPW), and *Pennise*–*tum purpureum* Schum No. 2 (PP). (**A**) Microbe relative abundance; (**B**) Microbe Venn diagram; (**C**) Firmicutes to Bacteroide–tes ratio; (**D**,**E**) Relative abundance of each microbe in the broilers’ cecum at the phylum (**D**) or genus (**E**) level; (**F**–**H**) Taxa heatmap of cecum microbes according to order (**F**), family (**G**) and genus (**H**). (**I**–**L**) Alpha diversity of 35–day–old broilers’ ileum microbes in each treatment; (**I**) Observed species, (**J**) Abundance-based coverage estimator (ACE), (**K**) Shannon index, and (**L**) Simpson index. Results are means of six samples obtained from the individual birds from the control and experimental groups (*p* < 0.05).

**Figure 4 animals-11-02550-f004:**
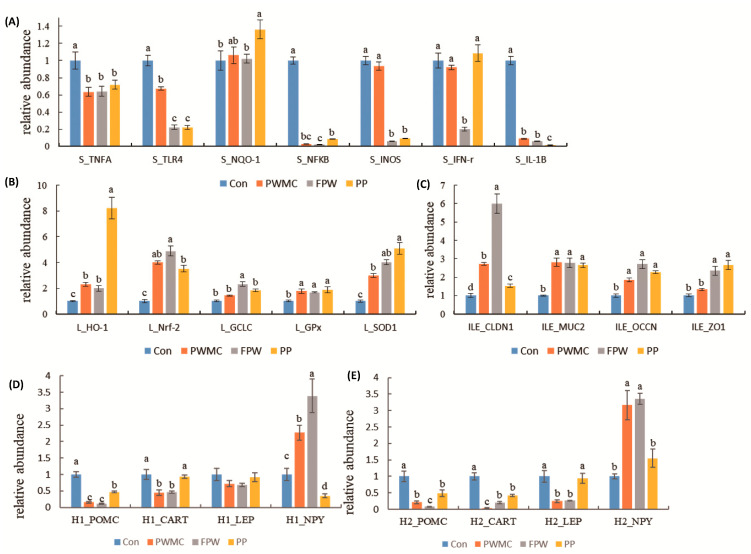
Relative abundance of mRNA expression in 35–day–old broilers given a corn–soybean basal diet (Con), *Pen*–*nisetum purpureum* Schum No. 2 waste mushroom compost (PWMC), *Saccharomyces cerevisiae* fermented PWMC (FPW), and *Pennisetum purpureum* Schum No. 2 (PP). (**A**) Inflammatory-related mRNA expression in broilers’ spleen; (**B**) Antiox–idant-related mRNA expression in broilers’ livers; (**C**) Gut barrier-related mRNA expression in broilers’ ileum; (**D**,**E**) Ap–petite-related mRNA expression in broilers’ hypothalamus before (H1, **D**) and after fasting (H2, **E**). Results are means of six samples obtained from the individual birds from the control and experimental groups. ^a–c^ Means within the same rows with distinct superscript letters are significantly different (*p* < 0.05).

**Table 1 animals-11-02550-t001:** Composition and calculated analysis (g/kg, as fed) of the basal and experimental diets for broilers, for both the starter diet (1–21 days) and the finisher diet (22–35 days) ^1^.

Ingredients	Starter (1–21 Days)				Finisher (22–35 Days)			
	Con	PWMC	FPW	PP	Con	PWMC	FPW	PP
	----------------------------------------------g/kg---------------------------------------------							
Yellow corn	519.9	452.0	457.5	456.4	571.6	535.9	537.3	537.1
Soybean meal (CP 44.0%)	369.4	307.1	302.5	304.9	246.9	236.6	234.2	236.1
Full fat soybean meal	0.00	80.00	80.00	80.00	50.00	48.90	49.00	48.99
Fish meal (CP 65.0%)	30.00	30.00	30.00	30.00	50.00	50.00	50.00	50.00
Soybean oil	40.58	40.83	39.86	38.54	41.39	38.49	39.33	37.70
Calcium carbonate	16.50	16.50	16.50	16.50	16.50	16.50	16.50	16.50
Monocalcium phosphate	11.20	11.20	11.20	11.20	11.20	11.20	11.20	11.20
NaCl	3.9	3.9	3.9	3.9	3.9	3.9	3.9	3.9
_L_-Lysine-HCl	3.7	3.7	3.7	3.7	3.7	3.7	3.7	3.7
_DL_-Methionine	2.0	2.0	2.0	2.0	2.0	2.0	2.0	2.0
Choline-Cl	0.8	0.8	0.8	0.8	0.8	0.8	0.8	0.8
Vitamin premix ^2^	1.0	1.0	1.0	1.0	1.0	1.0	1.0	1.0
Mineral premix ^3^	1.0	1.0	1.0	1.0	1.0	1.0	1.0	1.0
PWMC	0	50	0	0	0	50	0	0
FPW	0	0	50	0	0	0	50	0
PP	0	0	0	50	0	0	0	50
Total	1000	1000	1000	1000	1000	1000	1000	1000
Calculated nutrient value								
Dry matter,%	88	89	89	88	88	91	90	90
Crude protein,%DM	23.0	23.0	23.0	23.0	21.0	20.9	21.0	21.0
Calcium,%DM	1.11	1.12	1.11	1.13	1.21	1.22	1.22	1.23
Total Phosphorus,%DM	0.68	0.72	0.72	0.69	0.72	0.76	0.76	0.72
Available Phosphorus,%DM	0.46	0.48	0.48	0.46	0.52	0.54	0.54	0.51
Methionine + Cystein,%DM	0.92	0.93	0.93	0.90	0.89	0.90	0.90	0.87
ME, kcal/kg DM	3050	3050	3050	3050	3175	3175	3175	3175
Chemical analysis value								
Dry matter,%	88.7	89.2	89.8	88.5	89.3	89.1	90.3	90.2
Crude protein,%DM	23.1	23.1	23.0	23.0	21.0	21.0	21.0	21.0
Crude fat,%DM Crude fiber,%DM	6.85	7.94	7.86	7.88	7.96	7.66	7.73	7.72
	3.60	6.42	6.14	6.78	3.11	6.03	5.74	6.39

^1^ Con: basal diet, PWMC: *Pennisetum purpureum Schum* No.2 waste mushroom compost, FPW: *Saccharomyces cerevisiae* fermented PWMC, PP: *Pennisetum purpureum Schum* No.2. ^2^ Vitamin (premix content per kg diet): vit. A, 15 000 IU; vit. D3, 3000 IU; vit. E, 30 mg; vit. K3, 4 mg; thiamine, 3 mg; riboflavin, 8 mg; pyridoxine, 5 mg; vitamin B12, 25 μg; Ca-pantothenate, 19 mg; niacin, 50 mg; folic acid, 1.5 mg; and biotin, 60 μg. ^3^ Mineral (premix content per kg diet): Co (CoCO3), 0.255 mg; Cu (CuSO_4_ 5H_2_O), 10.8 mg; Fe (FeSO_4_ H_2_O), 90 mg; Mn (MnSO_4_ H_2_O), 90 mg; Zn (ZnO), 68.4 mg; Se (Na_2_SeO_3_), 0.18 mg.

**Table 2 animals-11-02550-t002:** The selected primer sequences of the Gallus gallus genes according to GenBank.

Gene Name ^1^	Primer Sequence	Accession No.	Size (bp)
*ß-actin*	F: 5′-CTGGCACCTAGCACAATGAA-3′ R: 5′-ACATCTGCTGGAAGGTGGAC-3′	X00182.1	109
*TNF*	F: 5′-TGTGTATGTGCAGCAACCCGTAGT-3′ R: 5′-GGCATTGCAATTTGGACAGAAGT-3′	NM 204267	229
*TLR4*	F: 5′-TGCACAGGACAGAACATCTCTGGA-3′ R: 5′-AGCTCCTGCAGGGTATTCAAGTGT-3′	NM_001030693	347
*NQO-1*	F: 5′-AAGAAGATTGAAGCGGCTGA-3′ R: 5′-GCATGGCTTTCTTCTTCTGG-3′	NM_001277619.1	166
*NFκB*	F: 5′- CCAGGTTGCCATCGTGTTCC-3′ R: 5′- GCGTGCGTTTGCGCTTCT-3′	D13719.1	179
*iNOS*	F: 5′-TACTGCGTGTCCTTTCAACG-3′ R: 5′-CCCATTCTTCTTCCAACCTC-3′	U46504	108
*IFN-γ*	F: 5′-CTCCCGATGAACGACTTGAG-3′ R: 5′-CTGAGACTGGCTCCTTTTCC-3′	Y07922	111
*IL-1ß*	F: 5′-GCTCTACATGTCGTGTGTGATGAG-3′ R: 5′-TGTCGATGTCCCGCATGA-3′	NM_204524	80
*HO-1*	F: 5′-GGTCCCGAATGAATGCCCTTG-3′ R: 5′-ACCGTTCTCCTGGCTCTTGG-3′	NM_205344.1	137
*Nrf-2*	F: 5′-GGAAGAAGGTGCGTTTCGGAGC-3′ R: 5′-GGGCAAGGCAGATCTCTTCCAA-3′	NM_205117.1	171
*GCLC*	F: 5′-CAGCACCCAGACTACAAGCA-3′ R: 5′-CTACCCCCAACAGTTCTGGA-3′	XM_419910.3	118
*GPX7*	F: 5′- CAGCAAGAACCAGACACCAA-3′ R: 5′- CCAGGTTGGTTCTTCTCCAG-3′	NM_001163245.1	156
*SOD-1*	F: 5′- ATTACCGGCTTGTCTGATGG-3′ R: 5′- CCTCCCTTTGCAGTCACATT-3′	NM_205064.155	173
*Claudin-1*	F: 5′-GGAGGATGACCAGGTGAAGA-3′ R: 5′-TCTGGTGTTAACGGGTGTGA-3′	NM_001013611.2	149
*MUC-2*	F: 5′-GCTACAGGATCTGCCTTTGC-3′ R: 5′-AATGGGCCCTCTGAGTTTTT-3′	JX284122.1	152
*Occludin*	F: 5′-GTCTGTGGGTTCCTCATCGT-3′ R: 5′-GTTCTTCACCCACTCCTCCA-3′	NM_205128.1	178
*ZO-1*	F: 5′-AGGTGAAGTGTTTCGGGTTG-3′ R: 5′-CCTCCTGCTGTCTTTGGAAG-3′	XM_015278975.1	145
*POMC*	5′-AAGGCGAGGAGGAAAAGAAG-3′ 5′-CCTTCTTGTAGGCGCTTTTG-3′	XM_015285103.2	168
*CART*	5′- CCTCGTGCAACTCCTTTCTT-3′ 5′- TTTCCTGAACGGACGAAAAC-3′	XM_003643097.4	173
*LEP*	5′-AGCAACGATTGAGGCGATT-3′ 5′-AGCAGCTCCTTCAACTCAGG-3′	LN794246.1	208
*NPY*	5′-CCTCATCACCAGGCAGAGAT-3′ 5′-CACTGGGAATGACGCTATGA-3′	NM_205473.1	212

^1^*TNF*: Tumor necrosis factor; *TLR4:* Toll-like receptor 4; *NQO-1*: NADPH dehydrogenase 1; *NFκB*: Nuclear factor kappa B p 65; *iNOS*: Inducible nitric oxide synthases; *IFN-γ*: Interferon-γ; *IL-1ß*: Interleukin-1ß; *HO-1*: Heme oxygenase-1; *Nrf-2*: Nuclear factor erythroid 2–related factor 2; *GCLC*: Glutamate-cysteine ligase catalytic; *Gpx*: glutathione peroxidase 7; *SOD-1*: Superoxide dismutase-1; *MUC-2*: Mucin2; *ZO-1*: Zonula occludens 1; *POMC*: Pro-opiomelanocortin; *CART*: Cocaine and amphetamine-regulated transcript; *LEP*: Leptin; *NPY*: Neuropeptide Y.

**Table 3 animals-11-02550-t003:** The functional chemical composition analysis of *Pennisetum purpureum* Schum No.2 waste mushroom compost, *Saccharomyces cerevisiae* fermented PWMC, and *Pennisetum purpureum* Schum No.2.

Items ^1^	PWMC	FPW	PP
Functional component analysis			
Crude polysaccharide (Glucose mg/g DM)	85.0 ± 1.23	83.0 ± 1.10	87.9 ± 2.1
Total VFA (μmole/g)	30.8 ± 1.4	28.1 ± 0.6	12.2 ± 0.3
Phenol-like chemical analysis (μg/g)			
Gallic acid	114 ± 2.7	159 ± 2.7	127 ± 2.4
Gallocatechin	1035 ± 8.0	1404 ± 23.0	1245 ± 15
Epigallocatechin	1493 ± 14.0	1817 ± 23.0	1687 ± 17
Catechin	7.90 ± 0.6	169 ± 0.9	13 ± 0.65
Caffeic acid	113 ± 0.5	235 ± 1.5	127 ± 1.2
Epicatechin	230 ± 2.0	213 ± 9.7	231 ± 5.8
Epigallocatechin gallate	210 ± 0.1	216 ± 3.5	202 ± 2.2
Epicatechin gallate	78.1 ± 2.8	199 ± 1.9	102 ± 2.2
Catechin gallate	401 ± 5.1	534 ± 7.3	430 ± 6.2
Chemical analysis ^2^			
DM	94.6 ± 0.4	94.2 ± 0.5	94.6 ± 0.6
CP (% DM)	6.8 ± 0.35	10.0 ± 0.53	12.6 ± 0.43
EE (% DM)	N.D ^3^	N.D	N.D
Ash (% DM)	10.5 ± 0.4	15.5 ± 0.7	12.4 ± 0.8
NDF (% DM)	59.7 ± 0.8	54.4 ± 1.6	55.2 ± 0.9
ADF (% DM)	48.8 ± 2.8	34.3 ± 0.5	38.2 ± 1.7

^1^ PWMC: *Pennisetum purpureum* Schum No.2 waste mushroom compost; FPW: *Saccharomyces cerevisiae* fermented PWMC; PP: *Pennisetum purpureum* Schum No.2. ^2^ DM: Dry matter; CP: Crude protein; EE: Ether extract; NDF: Neutral detergent fiber; ADF: Acid detergent fiber. ^3^ Not detected. Results are means of five samples obtained from each product.

**Table 4 animals-11-02550-t004:** Serum characteristic of 35-day-old broilers in different treatments ^1^.

Items	Treatments				SEM ^2^	*p* Value
	Con	PWMC	FPW	PP		
GLU (mg/dL)	276 ^a^	236 ^b^	240 ^b^	227 ^b^	5.2	<0.0001
Protein						
SGOT (U/L)	332	251	248	246	24	0.0555
SGPT (U/L)	4.17	3.67	4.33	5.33	0.72	0.4432
T-P (g/dL)	3.4	3.18	3.1	3.13	0.08	0.0534
ALB (g/dL)	1.55 ^a^	1.47 ^ab^	1.4 ^b^	1.45 ^ab^	0.03	0.0432
GLO (g/dL)	1.85	1.72	1.7	1.68	0.06	0.1662
Alk-P (IU/L)	1086	1280	1366	1357	234	0.8185
Gpx (nmol/min/mL)	114.7	143.3	124.2	219.8	41.9	0.4080
SOD (mU/mL)	523.9 ^b^	539.2 ^b^	672.1 ^a^	620.7 ^a^	23.42	0.0009
TNF-α (pg/mL)	284.7 ^a^	145.4 ^b^	155.9 ^b^	195.8 ^b^	19.56	0.0019
Lipid						
CHOL (mg/dL)	125	121	112	120	4.1	0.1542
TG (mg/dL)	54.8 ^a^	40.5 ^b^	38.2 ^b^	40.2 ^b^	2.7	0.0008
HDL-C (mg/dL)	78	74.7	69.3	72.2	2.7	0.1562
LDL-C (mg/dL)	42.2	39.5	39.7	44.2	2.0	0.3415

^1^ Con: basal diet (corn–soybean); PWMC; FPW: 5% replacement of a soybean meal via FPW (*Saccharomyces cerevisiae* fermented PWMC); PP: 5% replacement of a soybean meal via PWMC (PWMC–*Pennisetum purpureum Schum* No.2 waste mushroom compost). ^2^ SEM: Standard error of means. ^a,b^ Means within the same rows with distinct superscript letters are significantly different (*p* < 0.05). Results are means of nine samples obtained from the individual birds of each control and experimental groups.

## Data Availability

The data presented in this study are available on request from the corresponding author.

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
