# Peer review of "Intestinal Microbiota, Anti-Inflammatory, and Anti-Oxidative Status of Broiler Chickens Fed Diets Containing Mushroom Waste Compost By-Products"

_animals, 2021, doi:10.3390/ani11092550_

Round 1
Reviewer 1 Report
The publication is interesting and the text is well written. However, I have a few methodological doubts and I am asking for a supplement.
- from how many individuals were tissues collected for RNA isolation?
- what were the conditions of the qPCR reaction (temperature profile)?
- as far as I know, the formula for calculating the relative expression of genes according to the Livak and Schmittgen 2001 methodology assumes a superscript in formula - however, the authors did not provide references according to which methodology they used to analyze
- what was the melting point? new starters were designed - with what design tool? have the primers been optimized?
- why was B-actin selected as the reference gene? Why were literature primer sequences optimized by other authors not used?
- what are the lengths of the amplification products for the designed primers? is the sequence on an exon-exon junction?
- statistical analysis - was the distribution consistent with the normal? Has the homogeneity of variance analysis been performed?
Author Response
The authors appreciate the comments from the reviewers. The manuscript has been revised in accordance with their requests. We have tried our best to take all comments into account, incorporating them into the revised manuscript as indicated in our responses to the reviewer.
(Revisions related to reviewer’s comments are shown in blue in the revised manuscript)
Responses to Reviewer I 's comments:
The publication is interesting and the text is well written. However, I have a few methodological doubts and I am asking for a supplement.
Response: We really appreciate the comments, and thank you for providing the useful suggestion.
- from how many individuals were tissues collected for RNA isolation?
Response: Many thanks for your kind review. We have added the description as “Briefly, total mRNA from each tissue (approximately 100 mg for individuals tissues) was isolated following the manufacturer’s protocol (SuperScript™ FirstStrand Synthesis System reagent, Invitrogen, USA) and purity was determined by the absorbance ratio of 260/280 nm.” - what were the conditions of the qPCR reaction (temperature profile)
Response: Thanks for the suggestion. We have added the description as “The PCR conditions were as follows: initial denaturation at 95°C for 20 seconds, 40 cycles of 95°C for 3 seconds, 55°C for 30 seconds, and 72°C for 30 seconds, and then final extension at 72°C for 5 minutes.” in the Total RNA isolation and qPCR - as far as I know, the formula for calculating the relative expression of genes according to the Livak and Schmittgen 2001 methodology assumes a superscript in formula - however, the authors did not provide references according to which methodology they used to analyze
Response: Thanks for providing the useful suggestion. We have added the reference and description as “The methodology was accorded to Schmittgen and Livak [28]” in the Total RNA isolation and qPCR
Ref:
[28] Schmittgen T.D.; Livak, K.J. Analyzing real-time PCR data by the comparative C(T) method. Nat. Protocols 2001. 3(6), 1101-1108. - what was the melting point? new starters were designed - with what design tool? have the primers been optimized?
Response: Many thanks for your kind review. The melting point of the genes as below table (A typical denaturation (melting) curve performed after a qPCR cycle using an intercalating dye, usually produces a unique peak in the graph of the negative fluorescence derivative versus temperature. This indicates that the amplified double-stranded DNA product is a single discrete species). The study primers were designed by Primer Express 3.0.1. Moreover, the primers were processed by OPC (Oligonucleotide Purification Cartridge). - Why was B-actin selected as the reference gene? Why were literature primer sequences optimized by other authors not used?
Response: Thanks for providing the useful suggestion.β-actin is often used as an internal control (gene) or housekeeping gene to normalize expression of target gene or mRNA levels between different samples (Li Z et al., 2010). However, the stability of β-actin gene had been proven by Kasper et al (2010). Moreover, β-actin has been used as housekeeping gene in other broiler chickens studies (Zhong et al., 2014; Mueller et al., 2012). Thanks again for valuable comments for our future study.
Ref:
(1) Kasper G, Ode A, Groothuis A, Glaeser J, Gaber T, Wilson CJ, Geissler S, Duda GN. 2010. Validation of β‐Actin used as endogenous control for gene expression analysis in Mechanobiology studies: amendments. Stem Cells. 28(3):633-634.
(2) Zhong X, Shi Y, Chen J, Xu J, Wang L, Beier RC, Hou X, Liu F. 2014. Polyphenol extracts from Punica granatum and Terminalia chebula are anti-inflammatory and increase the survival rate of chickens challenged with Escherichia coli. Biol Pharm Bull. 37(10):1575-1582.
(3) Mueller K, Blum NM, Kluge H, Mueller AS. 2012. Influence of broccoli extract and various essential oils on performance and expression of xenobiotic- and antioxidant enzymes in broiler chickens. Br J Nutr. 108:588–602.
(4) Li Z, Yang L, Wang J, Shi W, Pawar RA, Liu Y, Xu C, Cong W, Hu Q, Lu T, Xia F, Guo W, Zhao M, Zhang Y. 2010. Beta-Actin is a useful internal control for tissue-specific gene expression studies using quantitative real-time PCR in the half-smooth tongue sole Cynoglossus semilaevis challenged with LPS or Vibrioanguillarum. Fish Shellfish Immunol. 29(1):89-93. doi: 10.1016/j.fsi.2010.02.021. - What are the lengths of the amplification products for the designed primers? is the sequence on an exon-exon junction?
Response: Thanks for the suggestion. We have added the lengths of the amplification products for the designed primers in Table 2 (as below). And, yes, the sequences on an exon-exon junction. - Statistical analysis - was the distribution consistent with the normal? Has the homogeneity of variance analysis been performed?
Response: We really appreciate the comments. The data was normally was tested by normal distribution test. The homogeneity of the variances among treatments was confirmed using the UNIVARIATE procedure and this procedure was also used to identify outliers, but no outliers were observed. We have added the description as “Variable of serum, intestine and gut microbiota was tested by normal distribution and homogeneous.” In the 5 Total RNA isolation, qPCR and sequencing section.

Reviewer 2 Report
In this experiment, the author analyzed the Serum characteristic of broilers, gut microbiota composition, and mRNA expression in liver, spleen, and ileum of broilers and other indicators under the four treatments to explore the effects of high-fiber agricultural and sideline products on broiler microbiology under different treatments. And evaluate its anti-inflammatory and antioxidant capacity in broiler chickens to find the appropriate replacement amount in broiler feed. In particular, the research on the influence of feed types on the composition of gut microbiota composition and the utilization of Mushroom wastes compost have high production and utilization value.
However, there are still some problems and errors that need to be corrected.
- In the abstract part, the author gave a simple description of the results, but the conclusive summary of the antioxidative ability or inflammatory response of the chicken of the mushroom wastes and compost is missing.
- Line 45-48: Please add a reference.
- Line 64-67: There seems to be no logical relationship between these two sentences, there is no causal relationship, please rewrite.
- Line 72-77: Many of the sources cited here are from the same reference. Please refine this section, or cite other references. In addition, such as [7], [9] etc. have been quoted repeatedly in the paper, which may make the paper less convincing.
- In the materials and methods, the detection method of the type and abundance of intestinal microorganisms is not stated.
- Line 116: Please state the full name where the first abbreviation appears.
- Line 168: How to determine 5% PWMC as the best replacement quantity? why it's 5%? Has the author tried any other ratios?
- Line 219: How many biological and technical replicates were used? How many microliter systems are used for q-PCR?
- Line 386-396: The fasting treatment mentioned in the article is not reflected in the animal experiment part.
Minor points:
- Line 13: “Mushroom waste” should be modified to “Mushroom wastes”.
- Line 101: Change “-20℃refrigerators” to “-20℃ refrigerators”.
- It is recommended to improve the clarity of Figure 2.
- Line 214: Change “the” to “The”.
- Line 231: Please modify it as superscript.
- The format of Table 2 and Table 4 are messy, please modify it.
- Please delete the "t" in front of Pleurotus eryngii.
Author Response
Responses to Reviewer II 's comments:
In this experiment, the author analyzed the Serum characteristic of broilers, gut microbiota composition, and mRNA expression in liver, spleen, and ileum of broilers and other indicators under the four treatments to explore the effects of high-fiber agricultural and sideline products on broiler microbiology under different treatments. And evaluate its anti-inflammatory and antioxidant capacity in broiler chickens to find the appropriate replacement amount in broiler feed. In particular, the research on the influence of feed types on the composition of gut microbiota composition and the utilization of Mushroom wastes compost have high production and utilization value.
Response: We really appreciate the comments, and thank you for providing the useful suggestion.
However, there are still some problems and errors that need to be corrected.
- In the abstract part, the author gave a simple description of the results, but the conclusive summary of the antioxidative ability or inflammatory response of the chicken of the mushroom wastes and compost is missing.
Response: Thanks for the suggestion. We newly added the description as ”Overall, high fiber feeds (PWMC, FPW, and PP) could enhance broilers health by inducing antioxidant capacities and decrease inflammatory response in compare to the control. According to the results, 5 % PWMC replacement was recommended in diet”. - Line 45-48: Please add a reference.
Response: Thanks for the suggestion. We have modified and added a reference as “Carbohydrates, one of the main sources of energy for animals, can roughly divide into two broad categories. First category is the source of energy for animals, such as starch, glucose, and sucrose, which are decomposed by enzymes produced by animals. Second category is dietary fiber that cannot digest by the animal itself [6]”.
Ref:
[6] Buxton, D.R.; Redfearn, D.D. Plant limitations to fiber digestion and utilization. J. Nutr. 1997, 127(5), 8145-8185. https://doi.org/10.1093/jn/127.56.814S
- Line 64-67: There seems to be no logical relationship between these two sentences, there is no causal relationship, please rewrite.
Response: Thanks for the suggestion. We have corrected the description as ” Nevertheless, there are more and more studies have use the agricultural by-products as the source of animal feed ingredients, and the results showed no reduce the production performances of animals and even could help improve animal health [7,8]. The season be suggested the high-fiber feed ingredients help to strengthen the intestinal barrier and antioxidant capacity of animals, and also reduce inflammation [4]. Those additional antioxidant capacity can help in conditions of sudden oxidative stress, whether the negative factors came from the environments or pathogenic bacteria [9]. - Line 72-77: Many of the sources cited here are from the same reference. Please refine this section, or cite other references. In addition, such as [7], [9] etc. have been quoted repeatedly in the paper, which may make the paper less convincing.
Response: Thanks for the suggestion. We have corrected and added the 7 references as “When animals are subjected to environmental stress, such as pathogens infection or heat stress, stimulation causes an increases oxidative stress and inflammatory [12,13]. Stimulated by pathogenic bacteria, the animal initiates an immune response leading to a cytokine storm. However, excessive inflammation can reduce animal performance and even lead to death [14,15]. On the other hand, when the oxidative pressure is high, animals are not able to eliminate the damage caused by free radicals to cells and/or organs [16]. Among them, the antioxidant system is mainly regulated by the liver, so the antioxidant capacity is also related to liver performance in animals [16-18].”
[12] Huang, C.M.; Lee, T.T. Immunomodulatory effects of phytogenics in chickens and pigs — A review. Asian-Aust. J. Anim. Sci. 2018, 31(5), 617-627.
[13] Diaz-Sanchez, S.; D'Souza, D.; Biswas, D.; Hanning, I.; Botanical alternatives to antibiotics for use in organic poultry production. Poult. Sci. 2015, 94(6), 1419-1430. https://doi.org/10.3382/ps/pev014
[14] Lin, W. C.; Lee, T.T. Effects of Laetiporus sulphureus-fermented wheat bran on growth performance, intestinal microbiota and digesta characteristics in broiler chickens. Animals. 2020, 10(9), 1457. doi.org/10.3390/ani10091457
[15] Kermanshahi, H.; Shakouri, M.D.; Daneshmand, A. Effects of non-starch polysaccharides in semi-purified diets on performance, serum metabolites, gastrointestinal morphology, and microbial population of male broiler chickens. Livest. Sci. 2018, 214, 93-97.
[16] Lee, M.T.; Lin, W.C.; Lee, T.T. Antioxidant capacity of phytochemicals and their potential effects on oxidative status in animals — A review. Asian-Aust. J. Anim. Sci. 2017, 30(3), 299-308.
[17] Lin, W.C.; Lee, T.T. Laetiporus sulphureus fermented wheat bran enhanced the antioxidant status, intestinal tight junction, and morphology of broiler chickens. Animals. 2021, 11(1):149 DOI:10.3390/ani11010149.
[18] Chen, L.W,; Chuang, W.Y.; Hsieh, Y.C.; Lin, H.H.; Lin, W.C.; Lin, L.J.; Chang, S.C.; Lee, T.T.. Effects of dietary supplementation with Taiwanese tea byproducts and probiotics on growth performance, lipid metabolism, and the immune response in red feather native chickens. Anim. Biosci. 2021. 34(3), 393-404. doi: 10.5713/ajas.20.0223.
- In the materials and methods, the detection method of the type and abundance of intestinal microorganisms is not stated.
Response: Thanks for the suggestion. We have added the description as ”After sequencing, whole tags were assembled using the UCHIME algorithm to detect chimera sequences; the chimera sequences were removed before the effective tags were obtained. Sequence analysis was performed using Uparse software (Uparse 135 v7.0.1001; http://drive5.com/uparse/). Sequences with ≥97% similarity were assigned to the same operational taxonomic units (OTUs). A representative sequence of each OTU was selected for further annotation. Alpha diversity was applied to analyze the complexity of the species diversity for a sample by using six indices: Observed OTUs, Shannon, Simpson, Chao1, ACE, and PD whole tree. All the indices of our samples were calculated using Quantitative Insights Into Microbial Ecology (QIIME, v1.9.1). To evaluate differences in samples with respect to species complexity, beta diversity analysis on weighted unifrac were conducted using QIIME software (v1.9.1). Principal Co-ordinates analysis (PCoA) was performed at the genus level. LEfSe (Linear discriminant analysis effect size) was performed to detect differential abundant taxa across groups using the default parameters.” in the 5 Total RNA isolation, qPCR and sequencing section. - Line 116: Please state the full name where the first abbreviation appears.
Response: Thanks for the suggestion. We have corrected the description it. - Line 168: How to determine 5% PWMC as the best replacement quantity? why it's 5%? Has the author tried any other ratios?
Response: Thanks for the suggestion. Mainly from the our previous similar study using Laetiporus sulphureus (LS, a kind of medicinal fungal) fermented wheat bran to evaluate the growth performance, intestinal microbiota and digesta characteristics in broiler chickens (Lin and Lee, 2020). The results showed 5% LS supplementation can improve growth performance as well as exert immunomodulatory functions to dampen potential environment-induced inflammation, while 5% LS supplementation had a more ideal outcome. The crude fiber is similar (42.8% wheat bran vs1% PWMC), at the same time, they have both mycelium and abundant functional secondary metabolites in the fermented feeds/products (Lin and Lee, 2020; Ajila et al., 2012).
Ref:
Lin, W. C.; Lee, T.T. Effects of Laetiporus sulphureus-fermented wheat bran on growth performance, intestinal microbiota and digesta characteristics in broiler chickens. Animals. 2020, 10(9), 1457. doi.org/10.3390/ani10091457
Ajila, C.M.; Brar , S.K.; Verma, M.; Tyagi, R.D.; Godbout, S.; Valéro, J.R. Bio-processing of agro-byproducts to animal feed. Critical Reviews in Biotechnology 2012, 1-19. - Line 219: How many biological and technical replicates were used? How many microliter systems are used for q-PCR?
Response: Thanks for the suggestion. We collected from 6 broilers of each groups and replicated 6 times. And, 20 μl volume was used for q-PCR each time. We have added the description as “The mRNA of 35-day-old broilers were collected from 6 broilers of each groups and isolated from the liver (for anti-oxidant capacities measurement), spleen (for anti-inflammatory capacities measurement), ileum (for tight junction expression measurement), and hypothalamus (for appetite regulation measurement), respectively”. - Line 386-396: The fasting treatment mentioned in the article is not reflected in the animal experiment part.
Response: Thanks for the suggestion. We have corrected and added the description as “The mRNA of 35-day-old broilers were collected from 6 broilers of each groups and isolated from the liver (for anti-oxidant capacities measurement), spleen (for anti-inflammatory capacities measurement), ileum (for tight junction expression measurement), and hypothalamus (for appetite regulation measurement), respectively. Among them, the hypothalamus was removed from boilers before (H1) and after (H2) fasting for 24 h. During fasting, the birds provide with water spray and ventilation.”
Minor points:
- Line 13: “Mushroom waste” should be modified to “Mushroom wastes”.
Response: Thanks for the suggestion. We have corrected it. - Line 101: Change “-20℃refrigerators” to “-20℃ refrigerators”.
Response: Thanks for the suggestion. We have corrected it. - It is recommended to improve the clarity of Figure 2.
Response: Thanks for the suggestion. We have improve the clarity of Figure 2. - Line 214: Change “the” to “The”.
Response: Thanks for the suggestion. We have corrected it. - Line 231: Please modify it as superscript.
Response: Thanks for the suggestion. We have corrected it. - The format of Table 2 and Table 4 are messy, please modify it.
Response: Thanks for the suggestion. We have corrected the format of Table 2 and Table 4. - Please delete the "t" in front of Pleurotus eryngii.
Response: Thanks for the suggestion. Thanks for the suggestion. We have corrected it.

Reviewer 3 Report
The article entitled “Intestinal microbiota, anti-inflammatory, and antioxidative status of broiler chickens fed diets containing mushroom wastes compost byproducts” has been reviewed.
Overall, the article gives the sense that has been written frow two different authors. More specifically, the abstract and the introduction are quite weak and hard to follow while after these sections the overall layout is satisfactorily. For this reason, I would suggest to rewrite the simply summary and the introduction. Nevertheless, this work depicts a nice synergy of different protocols and methods resulting in interesting results. I would recommend its acceptance provided that my concerns will be addressed.
Specific comments:
English and syntax of simple summary are quite weak. Please rewrite.
Abstract: Line 30: Talk in past and provide results in abstract introduction. This is wrong.
Line 34: The mRNA levels or relative expression.
Line 35. Were increased.
Lines 49-50: The term “production-products” is repeated 3 times in a row.
The introduction is weak and does not include the up to now evidence in the topic. I suggest to rewrite the introduction leading straightforward to the objective of the present study.
Crude fiber should be also included in table 1.
2.3. Blood for enzymes activity should be stored in -80oC
Using only one housekeeping gene is quite risky aiming to obtain dependable results.
Please add software for primers design.
There are a lot of mistakes in genes names in table 2. For example, there is no TNFa (as gene anymore), it is TNF or LITAF, additionally you should clarify the GPX as well.
Author Response
Responses to Reviewer III 's comments:
The article entitled “Intestinal microbiota, anti-inflammatory, and antioxidative status of broiler chickens fed diets containing mushroom wastes compost byproducts” has been reviewed.
Overall, the article gives the sense that has been written from two different authors. More specifically, the abstract and the introduction are quite weak and hard to follow while after these sections the overall layout is satisfactorily. For this reason, I would suggest to rewrite the simply summary and the introduction. Nevertheless, this work depicts a nice synergy of different protocols and methods resulting in interesting results. I would recommend its acceptance provided that my concerns will be addressed.
Response: We really appreciate the comments, and thank you for providing the useful suggestion. We have rewrite and added the description (including added references) in the Simply Summary and Introduction section (as below).
Specific comments:
- English and syntax of simple summary are quite weak. Please rewrite.
Response: Thanks for the suggestion. We have rewrite the simple summary as “This study investigated the effects of supplementation Pennisetum purpureum waste mushroom compost (PWMC) on microbiota, antioxidant capacities and inflammatory response characteristics in broiler chickens. Results showed that 5 % PWMC replacement feeds could enhance chickens health by maintaining intestional microtiota balance, improving antioxidant capacities and decrease inflammatory response. Among them could further increase the appetite of broilers, thereby improving the growth performances. Furthermore, the number of Lactobacillus increases in intestinal tracts. High-fiber mushroom wastes compost could effectively increase mRNA expression of appetite-related genes in broilers. Broilers gut barrier function could also increase and the number of Turicibacter cecum decreased. It is concluded that 5 % PWMC could enhance intestine health, therefore, is commended in broiler’s diet.” and asked the professional proofreader to make the proof for the better English quality of this revised manuscripts. - Abstract: Line 30: Talk in past and provide results in abstract introduction. This is wrong.
Response: Thanks for the suggestion. We have rewrite the Abstract section as “This study investigated the effects of the mushroom wastes compost was the remaining medium after Pleurotus eryngii planting as a feed replacement on intestinal microbiota, anti-inflammatory, and anti- oxidative status of broiler chickens. Totally 240 broilers were used and allocated to 4 treatments, including corn-soybean basal diet, 5 % Pennisetum purpureum schum No.2 (PP), PWMC, and Saccharomyces cerevisiae fermented PWMC (FPW) replacement. The levels of glutathione peroxidase and superoxide dismutase mRNA as well as protein increased in liver and serum in chickens, respectively; mRNA levels of inflammation-related genes were also suppressed 2 to 10 times in all treatments compare to that in control. The tight junction and mucin were enhanced 2 to 10 times in all treatment groups compared to those in control, especially in the PWMC group. Nevertheless, the appetite-related mRNA levels were increased in PWMC and FPW groups by at least 2 times. In ileum and cecum, the Firmicutes/ Bacteroidetes ratios were decreased in PWMC, FPW, and PP groups in broilers. The Lactobacillaceae were increased mainly in PWMC and control groups in the ileum. Overall, high fiber feeds (PWMC, FPW, and PP) could enhance broilers health by improving antioxidant capacities and decrease inflammatory response in compare to the control. According to the results, 5 % PWMC replacement was recommended in diet.” - Line 34: The mRNA levels or relative expression.
Response: Thanks for the suggestion. We have corrected the description as “mRNA levels”. - Line 35. Were increased.
Response: Thanks for the suggestion. We have added the description as “were increased”. - Lines 49-50: The term “production-products” is repeated 3 times in a row.
Response: Thanks for the suggestion. We newly corrected the description as “Agricultural byproducts including wheat bran, crop hull, and straw, etc. are inevitable substances in the agricultural processing, but they are often not effectively used [1,2]. On the other hand, feed accounts for at least half of costs in the production of animal related products”. - The introduction is weak and does not include the up to now evidence in the topic. I suggest to rewrite the introduction leading straightforward to the objective of the present study.
Response: Thanks for the suggestion. We have rewrite the Introduction section as:
Agricultural byproducts including wheat bran, crop hull, and straw, etc. are inevitable substances in the agricultural processing, but they are often not effectively used [1,2]. On the other hand, feed accounts for at least half of costs in the production of animal related products [1,2]. However, with increasing awareness of environmental protection, agricultural wastes or byproducts are required to be properly treated [2,3,4]. In order to effectively reduce animal production costs and increase the value of agricultural by-products, previous studies have suggested the use of agricultural by-products such as mushroom wastes compost since it was rich in mycelium and contain abundant functional secondary metabolites, can improve antioxidant capacities and regulate the inflammatory response, as properly animal feed ingredients [3,4,5].
Carbohydrates, one of the main sources of energy for animals, can roughly divide into two broad categories. First category is the source of energy for animals, such as starch, glucose, and sucrose, which are decomposed by enzymes produced by animals. Second category is dietary fiber that cannot digest by the animal itself [6]. It can also distinguish between hard-to-digest fibers such as cellulose and chitin and other fibers according to whether the microorganism can be digested [5,6]. However, high-fiber agricultural by-products may reduce the performance of monogastric animals in traditional concepts [6], so this proposal has not been favored by animal producers for a long time. Nevertheless, there are more and more studies have use the agricultural by-products as the source of animal feed ingredients, and the results showed no reduce the production performances of animals and even could help improve animal health [7,8]. The season be suggested the high-fiber feed ingredients help to strengthen the intestinal barrier and antioxidant capacity of animals, and also reduce inflammation [4]. Those additional antioxidant capacity can help in conditions of sudden oxidative stress, whether the negative factors came from the environments or pathogenic bacteria [9]. High-fiber feed ingredients also help muscle formation and reduce adipose accumulation, thereby improving animal body composition [10]. In addition, the use of probiotics or enzymes to initially degrade agricultural by-products in vitro further increases the utilization of agricultural by-products [11].
When animals are subjected to environmental stress, such as pathogens infection or heat stress, stimulation causes an increases oxidative stress and inflammatory [12,13]. Stimulated by pathogenic bacteria, the animal initiates an immune response leading to a cytokine storm. However, excessive inflammation can reduce animal performance and even lead to death [14,15]. On the other hand, when the oxidative pressure is high, animals are not able to eliminate the damage caused by free radicals to cells and/or organs [16]. Among them, the antioxidant system is mainly regulated by the liver, so the antioxidant capacity is also related to liver performance in animals [16-18]. Fiber could reduce the inflammatory response and increase the antioxidant capacity of serum and liver, and enhance the growth of intestinal villi and health in animals [11]. Therefore, in the recent decade, the influence of gut microbiota composition has been paid increasing attention on animal health [19]. Previous studies have pointed out that gut microbiota composition is not only related to animal health, anti-inflammatory levels, and antioxidant capacity but can also affect animal appetite, circadian clock and mood [19]. The intestinal microbial composition can be changed rapidly and directionally by altering dietary patterns [7]. Under the combined effects of microorganisms and diet, the metabolome of animals may be changed [20]. Studies in mammals have shown that increasing the concentration of dietary fiber can promote the growth of Bacteroides, which is and reduce the number of Firmicutes in the gut [21]. Among them, Bacteroides are related to animals’ body maintenance, while Firmicutes are positively related to the degree of animal obesity [21]. In addition, increasing the dietary fiber concentration of animals also helps the growth of Bifidobacterium, which has a positive effect on animal health [21].
Although considerable research has been conducted on the effects of gut microbiota composition on mammalian health, only a few studies have been done on poultry. Different from mammals, poultry have different hormonal systems and physiological responses, so they need to be studied separately. Therefore, the purpose of this study was to investigate the effect of high fiber agriculture by-products on the broiler microbial phase under different treatments and to evaluate its anti-inflammatory and anti-oxidative capacity in broilers.
Addition ref:
[12] Huang, C.M.; Lee, T.T. Immunomodulatory effects of phytogenics in chickens and pigs — A review. Asian-Aust. J. Anim. Sci. 2018, 31(5), 617-627.
[13] Diaz-Sanchez, S.; D'Souza, D.; Biswas, D.; Hanning, I.; Botanical alternatives to antibiotics for use in organic poultry production. Poult. Sci. 2015, 94(6), 1419-1430. https://doi.org/10.3382/ps/pev014
[14] Lin, W. C.; Lee, T.T. Effects of Laetiporus sulphureus-fermented wheat bran on growth performance, intestinal microbiota and digesta characteristics in broiler chickens. Animals. 2020, 10(9), 1457. doi.org/10.3390/ani10091457
[15] Kermanshahi, H.; Shakouri, M.D.; Daneshmand, A. Effects of non-starch polysaccharides in semi-purified diets on performance, serum metabolites, gastrointestinal morphology, and microbial population of male broiler chickens. Livest. Sci. 2018, 214, 93-97.
[16] Lee, M.T.; Lin, W.C.; Lee, T.T. Antioxidant capacity of phytochemicals and their potential effects on oxidative status in animals — A review. Asian-Aust. J. Anim. Sci. 2017, 30(3), 299-308.
[17] Lin, W.C.; Lee, T.T. Laetiporus sulphureus fermented wheat bran enhanced the antioxidant status, intestinal tight junction, and morphology of broiler chickens. Animals. 2021, 11(1):149 DOI:10.3390/ani11010149.
[18] Chen, L.W,; Chuang, W.Y.; Hsieh, Y.C.; Lin, H.H.; Lin, W.C.; Lin, L.J.; Chang, S.C.; Lee, T.T.. Effects of dietary supplementation with Taiwanese tea byproducts and probiotics on growth performance, lipid metabolism, and the immune response in red feather native chickens. Anim. Biosci. 2021. 34(3), 393-404. doi: 10.5713/ajas.20.0223.
- Crude fiber should be also included in table 1.
Response: Thanks for the suggestion. We have added the Crude fiber in table - 3. Blood for enzymes activity should be stored in -80oC
Response: Thanks for the suggestion. We have corrected the description as “the serum was stored in -80 ℃ refrigerators before being analyzed.” - Using only one housekeeping gene is quite risky aiming to obtain dependable results.
Response: Thanks for the valuable suggestion. Currently we used theβ-actin is often used as an internal control (gene) or housekeeping gene to normalize expression of target gene or mRNA levels between different samples (Li et al., 2010). However, the stability of β-actin gene had been proven by Kasper et al. (2010). Moreover, β-actin has been used as housekeeping gene in other broiler chickens studies (Zhong et al., 2014; Mueller et al., 2012). However, we will keep the reminding from the reviewer for our nearly research. Thanks again.
Ref:
(1) Kasper G, Ode A, Groothuis A, Glaeser J, Gaber T, Wilson CJ, Geissler S, Duda GN. 2010. Validation of β‐Actin used as endogenous control for gene expression analysis in Mechanobiology studies: amendments. Stem Cells. 28(3):633-634.
(2) Zhong X, Shi Y, Chen J, Xu J, Wang L, Beier RC, Hou X, Liu F. 2014. Polyphenol extracts from Punica granatum and Terminalia chebula are anti-inflammatory and increase the survival rate of chickens challenged with Escherichia coli. Biol Pharm Bull. 37(10):1575-1582.
(3) Mueller K, Blum NM, Kluge H, Mueller AS. 2012. Influence of broccoli extract and various essential oils on performance and expression of xenobiotic- and antioxidant enzymes in broiler chickens. Br J Nutr. 108:588–602.
(4) Li Z, Yang L, Wang J, Shi W, Pawar RA, Liu Y, Xu C, Cong W, Hu Q, Lu T, Xia F, Guo W, Zhao M, Zhang Y. 2010. Beta-Actin is a useful internal control for tissue-specific gene expression studies using quantitative real-time PCR in the half-smooth tongue sole Cynoglossus semilaevis challenged with LPS or Vibrioanguillarum. Fish Shellfish Immunol. 29(1):89-93. doi: 10.1016/j.fsi.2010.02.021. - Please add software for primers design.
Response: Thanks for the suggestion. The primers were designed by Primer Express 3.0.1. - There are a lot of mistakes in genes names in table 2. For example, there is no TNFa (as gene anymore), it is TNF or LITAF, and additionally you should clarify the GPX as well.
Response: Thanks for the suggestion. We have corrected the description as “TNF-α” and “GPX7” genes, and checked again those genes names.

Round 2
Reviewer 3 Report
The manuscript has been significantly updated in many parts.
Please revise the gene and protein names. There are some mistakes still.
I.E. TNF is the gene but the protein is TNF-A. In the primers table TNF (gene) should be written.
Please revise all gene and protein names.
Author Response
Responses to Reviewer III 's comments:
The manuscript has been significantly updated in many parts.
Please revise the gene and protein names. There are some mistakes still.
I.E. TNF is the gene but the protein is TNF-A. In the primers table TNF (gene) should be written.
Please revise all gene and protein names.
Response: We really appreciate the comments, and thank you for providing the useful suggestion. We have corrected the “TNF is the gene” and “the protein is TNF-α” in the revised manuscript.
